# Phase Angle as a Potential Screening Tool in Adults with Metabolic Diseases in Clinical Practice: A Systematic Review

**DOI:** 10.3390/ijerph20021608

**Published:** 2023-01-16

**Authors:** Samantha Praget-Bracamontes, Rogelio González-Arellanes, Carlos A. Aguilar-Salinas, Alexandro J. Martagón

**Affiliations:** 1Unidad de Investigación de Enfermedades Metabólicas, Instituto Nacional de Ciencias Médicas y Nutrición Salvador Zubirán, Mexico City 14080, Mexico; 2The Institute for Obesity Research, Tecnologico de Monterrey, Mexico City 64700, Mexico; 3Departamento de Endocrinología y Metabolismo, Instituto Nacional de Ciencias Médicas y Nutrición Salvador Zubirán, Mexico City 14080, Mexico; 4Escuela de Medicina y Ciencias de la Salud, Tecnologico de Monterrey, Mexico City 64700, Mexico

**Keywords:** phase angle, metabolic diseases, nutritional assessment, clinical implications, screening tool, bioelectrical impedance analysis

## Abstract

Background: Phase angle (PhA) has been used as mortality prognostic, but there are no studies about its possible use as a screening tool. Therefore, an assessment of the possible utility of PhA in clinical practice is required. The aim of this systematic review was to explore all recent available evidence of PhA, and its possible utility as a screening tool in clinical practice in subjects with chronic metabolic diseases. Materials and Methods: This systematic review was performed and written as stated in the PRISMA 2020 guidelines. The search was conducted in PubMed, ScienceDirect and SciElo. In order to be considered eligible, within the entire search, only articles involving PhA and their utility in metabolic diseases were included. Results: PhA was associated with hyperuricemia and vitamin D deficiency in obese subjects, and decreased cardiovascular risk and malnutrition in hospitalized patients. Conclusion: PhA may be a potential screening tool in clinical practice to evaluate different biomarkers, cardiovascular risk, and nutritional diagnosis in metabolic diseases in adults.

## 1. Introduction

In the last three decades, a worldwide increase in the prevalence of chronic metabolic diseases and associated health conditions has been reported [1]. This epidemiological scenery requires identifying subjects at risk in a timely manner to provide appropriate treatment and prevent complications, especially when it comes to healthcare systems with financial limitations and limited resources [1,2]. In this sense, phase angle (PhA) is an indicator of membrane integrity function, and has been suggested as an important prognostic indicator of mortality for critically ill patients and in some aspects that may impact directly: inflammation, and functional disabilities [2,3,4]. In addition, Norman et al. [4] suggested that PhA could be used as a non-invasive and non-expensive screening tool to identify subjects with impaired nutritional status in clinical practice.

According to Iragorri and Spackman [5], several methodological issues must be considered in the cost-effectiveness analysis (CEA) of new screening tools, for example, diagnostic test accuracy (sensitivity and specificity) and modeling false positive and negative results, among others. However, data related to PhA as a screening tool are limited, and a CEA is not always feasible to obtain. Currently, there is no clear information on which biomarkers are associated with PhA, and whether there would be any utility or clinical relevance in chronic metabolic diseases and associated health conditions.

In order to establish an approach to the possible utility of PhA in clinical practice, the aim of this review was to explore all recent available evidence of PhA and its possible usefulness, association or implication in chronic metabolic diseases assessment.

## 2. Materials and Methods

The systematic review was performed according to the Preferred Reporting Items for Systematic Reviews and Meta-Analysis (PRISMA) statement (Figure 1) [6].

### 2.1. Information Sources and Search Strategy

The steps specified by PRISMA were carried out to implement the systematic review. A search was conducted from August to October 2022. The included electronic databases were PubMed, ScienceDirect and SciElo, from January 2017 to August 2022. No filters were applied for the study design.

As a search strategy, all articles which contained the following keywords, MeSH terms and Boolean operators, were considered: “Phase Angle” AND “Metabolic Diseases” OR “Nutritional and Metabolic Diseases” OR “Metabolic Syndrome” OR “Obesity” OR “Overweight” OR “Dyslipidemias” OR “Diabetes Mellitus” OR “Hypertension” OR “Atherosclerosis” OR “Heart Diseases” OR “Myocardial infarction” OR “Stroke” OR “Renal Insufficiency”.

### 2.2. Eligibility and Exclusion Criteria

The PICOS strategy was adopted as “P” (participants or problem) of any age and ethnic group, of both sexes with a metabolic disease; “I” (intervention) was considered as PhA measurement in subjects with metabolic diseases; “C” (comparison) was not applicable; “O” (outcomes) was the association or implication of PhA values in metabolic diseases; and “S” (studies) were related only to cross-sectional, cohort studies and clinical trials.

In order to be considered eligible, all studies were required to meet the following inclusion criteria: both sexes, written in English language, the study design of all studies were descriptive observational studies (cross-sectional, prospective and retrospective cohort studies) and clinical trials (multicenter trials). Within the entire search, only studies involving PhA and its utility in metabolic diseases were included. No restriction was applied in relation to sample size.

The exclusion criteria were: articles without full-text availability, opinion pieces, review articles, meta-analysis, editorial letters, records with insufficient information about the involvement of PhA and metabolic diseases, written in Spanish language, and/or did not mention the topic.

### 2.3. Study Selection and Data Extraction

The protocol for this article was published in the International Prospective Register of Systematic Reviews, PROSPERO, under the register CRD42022381045. Data collection was carried out, beginning in August 2022, and concluding in October 2022. The title and abstracts of publications were updated throughout the screening step, and duplicate research was eliminated. Thereafter, the full text was analyzed to determine its suitability to accomplish a systematic review. All studies were collected and summarized in an electronic database.

Both titles and abstracts were reviewed separately by the authors. The following information was extracted for each selected study: (a) first author name and year of publication, (b) sample characteristics, (c) study design, (d) main outcomes, (e) BIA device and PhA equation, and (f) scientific evidence.

### 2.4. Certainty of Evidence

The certainty of evidence across studies was assessed using the grading of recommendations, assessment, development and evaluations (GRADE) framework. This framework allows the development and presentation of summaries of evidence, and provides a systematic approach for clinical practice recommendations, summarized in four categories of evidence: “high” (the authors have a large amount of confidence that the true effect is similar to the estimated effect), “moderate” (the authors believe that the true effect is probably close to the estimated effect), “low” (the true effect might be markedly different from the estimated effect), and “very low” (the true effect is probably markedly different from the estimated effect) [7]. Discrepancies were resolved by the three authors through consensus. The GRADE approach takes the study limitations (Risk of Bias), inconsistency of results, indirectness of evidence, imprecision, and publication bias, into consideration.

## 3. Results

### 3.1. Search Results

Overall, the initial strategy search retrieved a total of 361 publications (Figure 1), of which 48 were removed as duplicates. Following this, 313 records were screened for titles, of which 294 were excluded because they did not satisfy inclusion criteria. In total, 16 relevant articles/abstracts were reviewed in detail for eligibility and were included for the purposes of this systematic review, namely, nine cross-sectional studies, six cohort studies and one clinical trial (Table 1).

### 3.2. Phase Angle as a Potential Screening Tool in Metabolic Diseases

#### 3.2.1. Overweight and Obesity

According to the included studies, Table 2 and Figure 2, show that PhA may be used as an assessment indicator for the presence of hyperuricemia and vitamin D (25(OH)D) levels in overweight and obese subjects. A low PhA (<5°) was associated with the presence of hyperuricemia (uric acid > 6 mg/dL and > 7 mg/dL in women and men, respectively) in obese subjects (body mass index (BMI) ≥ 35.0 to 64.0 kg/m^2^) [8]. On the other hand, the lowest values of PhA were significantly associated with obesity and of 25(OH)D deficiency (OR = 0.3, and 0.2 respectively; *p* < 0.001) [9]. It is important to note that PhA was significantly correlated with several nutritional parameters such as albumin, blood urea nitrogen (BUN), creatinine, uric acid, phosphorus and glucose (r = 0.37, 0.31, 0.50, 0.46, 0.20 and −0.22, respectively; *p* < 0.05) [10].

#### 3.2.2. Malnutrition

In hospitalized patients, PhA could be used as an additional resource to identify subjects at risk of malnutrition and functionality level. Fernandez-Jiménez et al. [11] established that PhA values of ≤5.4° and ≤5.3° in men and women, respectively, admitted to hospital for different causes, were significantly associated with malnutrition or malnutrition-related diseases, as sorted by the malnutrition universal screening tool (MUST) scale.

#### 3.2.3. Cardiovascular and Chronic Kidney Diseases

Regarding functional level, an observational cohort study by Abe et al. [12] reported an independent association between low PhA (men < 5.62° and women < 4.54°) and functional independence measure motor (FIM-motor) in patients with acute stroke. Additionally, in chronic kidney disease (CKD) subjects, low values of PhA were associated with sarcopenia presence [13].

In other studies, it was reported that diabetic CKD stage 5 (DMCKD5) subjects with PhA values less than 4.17° were reported to show GNRI-assessed malnutrition and low albumin levels (3.13 ± 0.52 g/dL) compared with those with PhA ≥ 4.17° [14,15]. Finally, mechanical ventilation time and cardiac operative risk (EuroSCORE) were inversely associated with PhA in patients for elective cardiac surgery. Silva et al. [16] reported that every one point increase in EuroSCORE reduced PhA by 0.22°, and every minute more of mechanical ventilation (MV) reduced PhA by 0.00015°.

### 3.3. Differences on PhA Values in Several Health Conditions

Studies reported that obese subjects had a higher PhA in comparison with eutrophic subjects (6.9 ± 0.9° vs. 6.5 ± 0.8°; *p* = 0.003) [17]. Table 1 summarizes data from a multicenter clinical study of 217 individuals, showing a decrease in PhA with respect to age in type 2 diabetes mellitus (T2D) subjects compared with the control group [18]. Furthermore, this decrease was exacerbated and evident in relation to disease duration in patients with T2D [18]. Similarly, it was reported that non-alcoholic fatty liver disease (NAFLD) subjects had a higher PhA in comparison with non-NAFLD subjects (5.53 ± 0.66° vs. 5.43 ± 0.60°; *p* = 0.04), and PhA was associated with the risk of NAFLD [19]. In end-stage CKD patients, those with PhA < 6° values had higher levels of biomarkers related to arterial stiffness, compared with patients with PhA > 6° values [23].

### 3.4. Factors That Influence PhA Values

PhA was reported to be modified by various factors such as body mass index (BMI), age and gender. According to Barrea et al. [9], for each unit increase in BMI, PhA decreased 0.54° in subjects with a wide range of BMI. Contrary to these findings, Fu et al. [20] reported that in overweight and obese subjects, as BMI increased, there was an increase in PhA of 0.006°. Similarly, these authors agreed that for each increase in one year of age in subjects with a wide range of BMI, PhA decreased 0.11° and 0.014° in overweight and obese subjects, respectively [9,20]. Regarding sex, it was determined that the difference between men and women in PhA was 0.32° and 0.629°, respectively [9]. In addition, Streb et al. [21] reported that for each increase in the percentage of body fat, PhA decreased 0.66°. Contrarily, PhA was inversely correlated with central adiposity [22] (Table 2).

## 4. Discussion

It has been reported that obesity involves an inflammatory process that contributes to cardiovascular risk, and one of the mechanisms described is the presence of high uric acid levels that contribute to oxidative stress [24]. In this sense, low levels of vitamin D are also involved in inflammatory processes, mainly in the regulation of cytokine release [25]. In clinical practice, it is not common to assess certain biomarkers, such as vitamin D levels. Therefore, having a potential screening tool to identify possible subjects at risk of hyperuricemia and/or vitamin D deficiency could improve assessment in the clinical scope. According to the results obtained, in subjects living with obesity, a low PhA (<5°) is an indicator of hyperuricemia [8]. Additionally, PhA values are associated with vitamin D levels. Subsequently, this should be confirmed through a blood test, so as to initiate appropriate and timely treatment to prevent cardiovascular events [9]. Therefore, PhA could be a useful tool to assess the status of those biomarkers in subjects with excess adiposity.

In this setting, prompting a screening tool requires, in the first instance, establishing an association between the variable to be measured and the prognostic variable. Although a correlation between PhA and various biomarkers (BUN, creatinine, uric acid, phosphorus, and glucose) had previously been reported in the literature [10], it had not been determined that PhA could predict or be useful as a screening tool for such indicators. In this sense, correlation analysis indicates the intensity of the relationship between variables but not the effect or influence of one on the other.

According to our findings, PhA could be useful to identify subjects at risk of malnutrition in patients admitted to hospital for different causes (≤5.4° and ≤5.3° in men and women, respectively) [11], and functionality impairment in patients with acute stroke (men < 5.62° and women < 4.54°) [12]. These findings suggest that in the hospitalization context, a consideration of PhA to identify subjects with malnutrition and promptly initiate a suitable nutritional therapy should be incorporated to prevent length of stay and associated complications of malnutrition [26].

Additionally, identifying subjects with functional impairment would allow the placement of appropriate medical–nutrition strategies to avoid physical dependence, which could improve the life quality of subjects [27,28]. It is important to note that malnutrition and functionality impairment have a negative impact on healthcare systems and in several scenarios that involve patients, owing to increased hospital expenditures, drug consumption and length of recovery time [29]. Therefore, incorporating PhA into first-line nutritional assessment in the hospital area could show meaningful results [26,27]. In this sense, it would not represent a replacement as part of some step within first-line clinical practice; it solely could be enforced as a non-invasive and non-expensive additional resource in the nutritional scope.

On the other hand, the literature has demonstrated that there are differences in the values of PhA that subjects present under several different health conditions (obesity, diabetes and NAFLD) compared with healthy subjects. As mentioned above, confirmed subjects with diseases present alterations at the tissular level, which is reflected in their PhA values [8,18,19]. Particularly, PhA is influenced by age, BMI and sex, due to the variations in total body water (TBW) content [9,30]. It is well established that PhA is based on the resistance and reactance of the tissues, with TBW being the main component that influences these parameters [31]. It has been reported that TBW decreases with age, increases with the presence of obesity, and women have more liters of TWB compared with men [30,31].

Some limitations should be noted. Most of the selected studies were cross-sectional studies, which limits the power of the results and the possibility of making causal conclusions. In addition, the data were collected using different methods and in different countries, which may have influenced the results, especially given the heterogeneity of health conditions and characteristics of population (sex, ethnic and age group, etc.). None of the studies had the main objective of evaluating or validating PhA as a screening tool; they were secondary findings. Another issue concerns the model, scales, and instruments used to screen participants and the appropriateness of sample selection (different BIA devices), for example, monofrequency and multifrequency BIA.

To date, it is not possible to use phase angle widely in the clinical setting as a potential screening tool, as it requires appropriate validation in different conditions (by sex, ethnic and age group) and association studies. It would be opportune to carry out more studies in this regard, to clearly establish its behavior and utility.

## 5. Conclusions

In clinical practice phase angle could be a potential screening tool to evaluate different biomarkers, decrease cardiovascular risk and nutritional diagnosis, mainly concerning hyperuricemia and vitamin D deficiency in obese subjects. Additionally, age, sex and BMI influence PhA values. Therefore, more studies are required in this regard to elucidate their association and behavior thereof.

## Figures and Tables

**Figure 1 ijerph-20-01608-f001:**
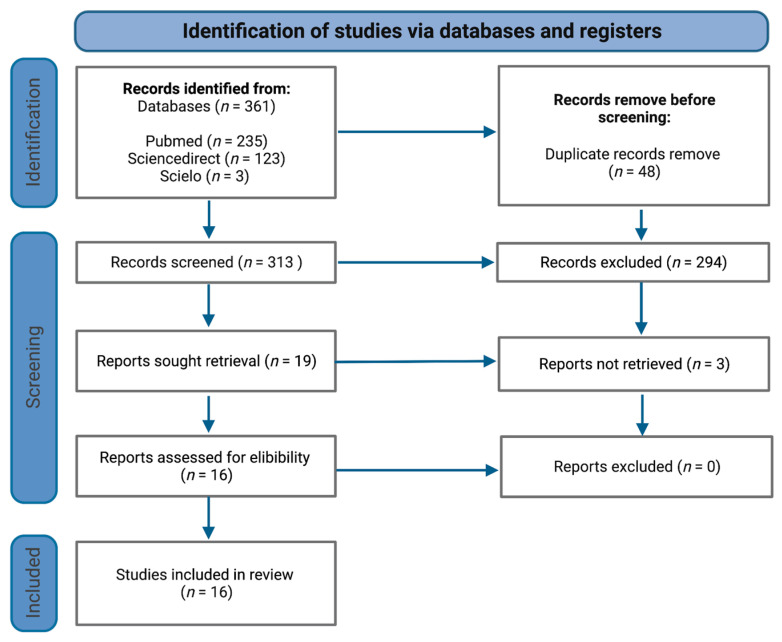
PRISMA flowchart of study selection process.

**Figure 2 ijerph-20-01608-f002:**
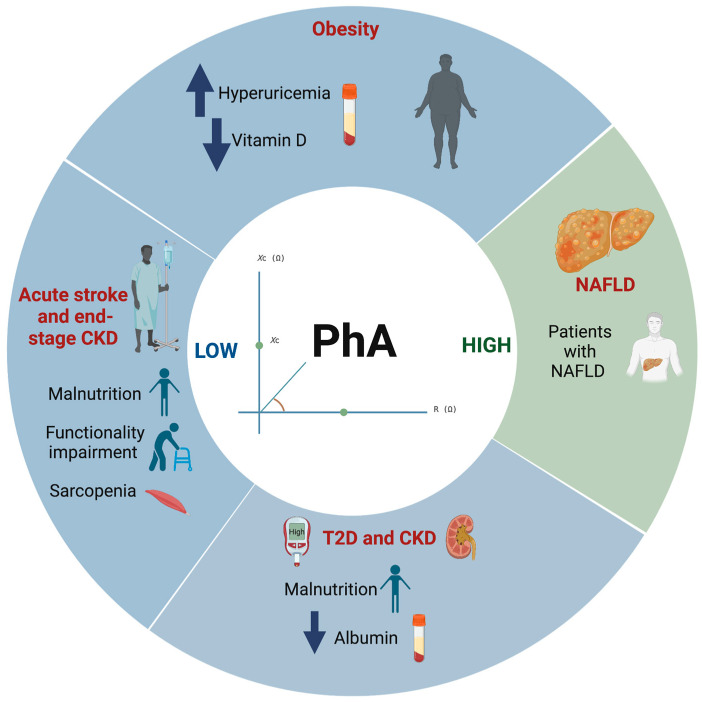
Graphic summary of reported results in this review, showing the relationship between PhA and metabolic diseases. Abbreviations: PhA: Phase angle; T2D: Type 2 Diabetes; CKD: Chronic kidney disease; NAFLD: Nonalcoholic fatty liver disease.

**Table 1 ijerph-20-01608-t001:** General description of included studies.

Authors and Year	Study Design	BIA Device and PhA Equation	Scientific Evidence
Curvello-Silva et al., 2020 [8]	Cross-sectional study	(a) DSM-BIA INBODY 720^®^; (b) PhA = arctan Xc/R.	C
Barrea et al., 2019 [9]	Cross-sectional study	(a) BIA 101 RJL; (b) PhA = arctangent Xc/R ((Xc/R) × (180/π)).	C
Shin et al., 2017 [10]	Retrospective observational study	(a) InBody S10, Biospace; (b) NR.	C
Fernández-Jiménez et al., 2022 [11]	Retrospective observational study	(a) BIA 101 RJL; (b) PhA = arctangent (Xc/R) × (180/π).	C
Abe et al., 2021 [12]	Observational cohort study	(a) InBody S10, Biospace; (b) PhA = arctangent (Xc/R).	B
Shin et al., 2022 [13]	Retrospective observational study	(a) InBody S10, Biospace; (b) PhA = arctangent (Xc/R) × (180/π).	B
Han et al., 2018 [14]	Cross-sectional study	(a) BIS BCM; (b) NR.	C
Han et al., 2019 [15]	Retrospective observational study	(a) BIS BCM; (b) NR.	B
Silva et al., 2018 [16]	Prospective cohort study	(a) Biodynamics 450^®^ analyzer; (b) NR.	B
de Oliveira-Filho et al., 2020 [17]	Cross-sectional study	(a) Tetrapolar device Quantum II, RJL system; (b) NR.	C
Jun et al., 2021 [18]	Multicenter clinical study	(a) InBody S10; (b) PhA*_f_*,_WB_ = sin^−1^((X*_f_*_,RA_ + X*_f_*_,TR_ + X*_f_*_,RL_)/(Z*_f_*_,RA_ + Z*_f_*_,TR_ + Z*_f_*_,RL_)).	B
Chen et al., 2020 [19]	Cross-sectional study	(a) InBody 770; (b) PhA = arctangent (Xc/R) × (180/π).	C
Fu et al., 2022 [20]	Observational retrospective study	(a) Inbody 770; (b) PhA = arctangent (Xc/R) × (180/π).	C
Streb et al., 2020 [21]	Cross-sectional study	(a) InBody^®^ 720 model; (b) PhA = arctangent (Xc/R) × (180/π).	C
Ferreira et al., 2018 [22]	Cross-sectional study	(a) Biodynamics BIA-450 body fat analyzer; (b) NR.	C
Sarmento-Dias et al., 2017 [23]	Cross-sectional study	(a) Multifrequency BIA using Inbody S10; (b) NR.	C

Abbreviations: PhA: phase angle, NR: not reported, Xc: reactance, R: resistance, BCM: Body Composition Monitor^TM^. Scientific evidence according to GRADE scale: High = A, Moderate = B, Low = C, and Very Low = D.

**Table 2 ijerph-20-01608-t002:** Main outcomes of included studies.

Authors and Year	Sample Characteristics (a) Size, *n* (b) Age, Years (c) BMI, kg/m^2^ (d) PhA, °	Main Outcomes
Curvello-Silva et al., 2020 [8]	(a) *n* = 141; (b) 37.9 ± 10.7 years; (c) ≥35.0 to 64.0 kg/m^2^; (d) 3.32 to 7.21°.	Association between low PhA (<5°) and presence of hyperuricemia, adjusted by waist circumference, dysglycemia, and arterial hypertension (*p* = 0.018).
Barrea et al., 2019 [9]	(a) *n* = 287 obese, 79 normal, and 89 overweight subjects; (b) 37.0 ± 11.0 years; (c) 34.0 ± 8.0 kg/m^2^; (d) 5.8 ± 0.8°.	BMI, sex and age were associated with PhA (β = −0.54, −0.32, −0.11, respectively; *p* ≤ 0.004). The lowest values of PhA were significantly associated with obesity (OR = 0.3) and 25(OH)D deficiency (OR = 0.2). The specific cut-off for 25(OH)D levels to predict the PhA above the median was >14 ng/mL (*p* < 0.001).
Shin et al., 2017 [10]	(a) *n* = 142 patients with end-stage CKD; (b) 64 ± 13 years; (c) 22.5 kg/m^2^; (d) 4.6 ± 1.0°.	A positive correlation of PhA with albumin, BUN, creatinine, uric acid, and phosphorus (r = 0.37, 0.31, 0.50, 0.46, and 0.20; *p* < 0.05). A negative correlation of PhA with glucose and TCO_2_ (r = −0.22 and −0.19; *p* = 0.009 and 0.025).
Fernández-Jiménez et al., 2022 [11]	(a) *n* = 570 patients admitted to hospital for different causes; (b) 65.0 years (IC: 53.0–74.0); (c) 24.9 kg/m^2^ (IC: 22.0–28.1); (d) 5.1 ° (IC: 4.1–6.1).	PhA cut-off for malnutrition diagnosis was 5.4°, 5.4° and 5.3°, in total sample, men and women, respectively.
Abe et al., 2021 [12]	(a) *n =* 129 patients with stroke; (b) 75.2 ± 12.9 years; (c) 22.8 ± 4.0 kg/m^2^; (d) 5.27 ± 1.1°.	Independent association between low PhA (<5.62° in men and <4.54° in women) and physical function (β = 0.201, *p* < 0.017), after adjustment.
Shin et al., 2022 [13]	(a) *n* = 149 patients with end-stage CKD; (b) 65.0 ± 11.0 years; (c) 25.3 ± 3.0 kg/m^2^; (d) 5.4 ± 1.1°.	Low PhA values were associated with the presence of sarcopenia, independent of age, sex, comorbidity index, eGFR, and uPCR (OR: 0.12; *p* = 0.001).
Han et al., 2018 [14]	(a) *n* = 160 patients with CKD stage 5; (b) 56.9 ± 9.9 years; (c) 24.7 ± 3.8 kg/m^2^; (d) 4.70 ± 1.29°.	Association between PhA and nutritional status (GNRI > 98 score, β = 0.152, *p* = 0.037).
Han et al., 2019 [15]	(a) *n* = 219 patients with diabetic CKD stage 5; (b) 60.3 ± 13.5 years; (c) 24.8 ± 4.0 kg/m^2^; (d) 4.31 ± 1.22°.	Albumin level (OR: 0.131; *p* < 0.001) was significantly associated with undernutrition (PhA < 4.17°) in the DMCKD5 group.
Silva et al., 2018 [16]	(a) *n* = 50 patients for elective cardiac surgery; (b) 62.8 ± 10.2 years; (c) NR; (d) Preoperative, 6.5° to 6.8°; hospital discharge, 5.9° to 6.3°; and postoperative, 5.8° to 6.2°.	The mechanical ventilation time and European system for cardiac operative risk (EuroSCORE) were inversely associated with PhA in all three assessments (*p* = 0.05).
de Oliveira-Filho et al., 2020 [17]	(a) *n* = 78 obese and 411 eutrophic subjects; (b) 16.2 ± 0.9 years; (c) Obese, 28.6 ± 3.4 kg/m^2^; eutrophic, 20.1 ± 2.2 kg/m^2^; (d) 6.9 ± 0.9°.	Obese subjects had a higher PhA in comparison with eutrophic subjects (6.9 ± 0.9° vs. 6.5 ± 0.8°; *p* = 0.003).
Jun et al., 2021 [18]	(a) *n* = 217 Korean adults; (b) ≥40 years; (c) 25.9 ± 3.9 kg/m^2^; (d) 5.18 to 5.98°.	Decrease in PhA with respect to age in DMT2 vs. control group. Decrease in PhA in patients with DMT2, and the changes were exacerbated over the disease duration.
Chen et al., 2020 [19]	(a) *n* = 271 non-NAFLD, and 682 NAFLD subjects; (b) 44.0 ± 9.7 years; (c) 28.4 ± 3.1 kg/m^2^; (d) 5.5 ± 0.65°.	NAFLD subjects had a higher PhA in comparison with non-NAFLD subjects (5.53 ± 0.66° vs. 5.43 ± 0.60°; *p* = 0.04). Association between PhA and the risk of NAFLD, after adjustment (OR = 1.40, *p* = 0.03).
Fu et al., 2022 [20]	(a) *n* = 1729 overweight and obese subjects; (b) 34.6 ± 10.7 years; (c) 33.7 ± 6.4 kg/m^2^; (d) 5.5 ± 0.6°.	BMI, sex, and age were associated with PhA (β = 0.006, 0.629, −0.014, respectively; *p* < 0.05).
Streb et al., 2020 [21]	(a) *n* = 69 obese subjects; (b) 34.6 ± 7.1 years; (c) 33.5 ± 2.8, kg/m^2^; (d) 5.8 ± 0.6°.	An increase of 1% point in body fat representing a reduction of 0.065° in PhA, after adjustment (*p* < 0.001).
Ferreira et al., 2018 [22]	(a) *n* = 86 renal transplant recipient subjects, and 86 subjects in the comparison group; (b) 45 to 70 years; (c) 25 to 35 kg/m^2^; (d) 6.1° (5.6 to 7.0).	PhA was inversely and significantly correlated with waist-to-height ratio and body shape index, r = −0.22, −0.21, respectively; *p* < 0.05.
Sarmento-Dias et al., 2017 [23]	(a) *n* = 61 patients with end-stage CKD; (b) 48 ± 13 years; (c) NR; (d) NR.	Low PhA (<6°) had higher CRP, AI, and SCS values, and lower serum albumin and fetuin-A levels compared with patients with high PhA (≥6°). Association between PhA and arterial stiffness, after adjustment (β = −0.266, *p* = 0.088).

Abbreviations: PhA: phase angle; BUN: blood urea nitrogen; BMI: body mass index; NR: not reported; CKD: chronic kidney disease; Xc: reactance; R: resistance; NAFLD: non-alcoholic fatty liver disease; T2D: type 2 diabetes; CRP: C-reactive protein; AI: augmentation index; SCS: simple vascular calcification score.

## Data Availability

The data presented in this study are available on request from the corresponding author.

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
