# Peer review of "Phase Angle as a Potential Screening Tool in Adults with Metabolic Diseases in Clinical Practice: A Systematic Review"

_ijerph, 2023, doi:10.3390/ijerph20021608_

Round 1
Reviewer 1 Report
Praget-Bracamontes et cols. presented a sistematic review that aimed to explore all recent evidence available of PhA and its possible utility as a screening tool in chronic metabolic diseases assessment. The follow of PRISMA statement contribute for warranty the quality of selection of studies. We congratulate the author for the paper but we recommend reanalysis for some improvements in some topics.
In the 1st line of Abstract the authors bring “no studies about possible use as a screening tool”. It is not clear for us why authors say that if they bring 16 studies that show association between variables and PhA and discuss its possible use as a screening tool. Maybe it could be better say the “no studies about the use as a precise screening tool”
We believe the Table 1 could have less information to give the reader a clearer sight to data. A possible solution could be using some abbreviations and adjust the configuration of information inside the cell.
Maybe the data analysis and discussion could bring more contributions if made clustered by age (adolescent, adults, elderly), perhaps dividing the table in 3 smaller ones. It is well described in the literature (and the authors bring this information) that PhA suffers influence by age because the cellularity changes with it. The review purpose is to clarify data about PhA use as a screening tool and this application is not the same for every age group.
Author Response
The authors thank the reviewers sincerely for their careful reading of the manuscript and their suggestions for improvement.

Reviewer 2 Report
Dear IJERPH,
The aim of this systematic review was to explore all recent evidence available of PhA and its possible utility as a screening tool in chronic metabolic diseases assessment. It is a robust research (i.e., systematic review) with an interesting research aim (i.e., applying phase angle as a potential screening tool for metabolic diseases), however some sections should be improved.
Please see the major revisions in attachment.
Good work!

Author Response

(The authors gave the same response as above.)

Round 2
Reviewer 2 Report
Dear author,
After the first round, it is noticeable that extensive revisions were made by the author to the article. Well done.
Best regards.